# A Novel Bio-Fertilizer Produced by Prickly Ash Seeds with Biochar Addition Induces Soil Suppressiveness against Black Shank Disease on Tobacco

Xifen Zhang [1,†], Yaochen Wang [1,2,†], Xiaobin Han [2,3,*], Jianyu Gou [2,3], Wei Li [4] and Chengsheng Zhang [1,*]

[1] Tobacco Research Institute of Chinese Academy of Agricultural Sciences, Qingdao 266101, China; zhangxifen0807@163.com (X.Z.); wangyaochen2020@163.com (Y.W.)
[2] Sinochem Agriculture Holdings, Beijing 100031, China; goujianyu2018@163.com
[3] Biological Organic Fertilizer Engineering Technology Center of China Tobacco, Zunyi Branch of Guizhou Tobacco Company, Zunyi 563100, China
[4] College of Life, Yangtze University, Jingzhou 434200, China; weli@yangtzeu.edu.cn
* Correspondence: hanxiaobin2011@163.com (X.H.); zhangchengsheng@caas.cn (C.Z.); Tel.: +86-0532-88702115 (C.Z.)
† These authors contributed equally to this work.

**Abstract:** A novel bio-fertilizer, produced from prickly ash seeds (PAS), *Bacillus subtilis* and biochar, was evaluated for its disease-preventing potential on tobacco black shank caused by *Phytophthora nicotianae*. The results showed that biochar promoted the growth of Tpb55 in PAS and increased the pH of the organic fertilizer. The final concentration of *B. subtilis* could reach $1.7 \times 10^{10}$ cfu g$^{-1}$ in the biological organic fertilizer (PBB) under the optimal medium under conditions of solid-state fermentation. PBB exhibited a strong fumigation effect on *P. nicotianae*, including inhibiting mycelium growth, reducing the disease severity and decreasing the pathogen population in rhizospheric soil. PBB treatment also could significantly increase the pH of acidified soil and improve soil nutrition content such as available K, alkali hydrolysable N and organic carbon. High-throughput pyrosequencing of 16S and 18S rRNA genes revealed that 4% PBB addition in soil had significant effects on the diversity and richness of fungi but not on that of bacteria. The microbial community structure was also shifted after PBB treatment. Some potentially beneficial microbes such as *Bacillus*, *Mucor*, *Cunninghamella*, *Chitinophaga* and *Phenylobacterium* were enriched, while potential pathogen *Fusarium* was significantly decreased. In conclusion, the agricultural waste PAS combined with biochar can replace soybean as a source for the production of biocontrol *B. subtilis* Tpb55, and the novel bio-fertilizer could effectively control tobacco black shank by pathogen inhibition, soil nutrient improvement and shifting the rhizomicrobial community.

**Keywords:** *Bacillus subtilis*; biochar; biological control; prickly ash seeds; rhizosphere microorganisms

## 1. Introduction

*Phytophthora nicotianae* van Breda de Haan is a typical pathogen of soil-borne diseases that causes great economic losses in many important crops, including tobacco, tomato and citrus [1,2]. Tobacco is one of the most important cash crops in the world, but it is seriously damaged by black shank disease caused by *P. nicotianae*. Currently, amide pesticides such as metalaxyl and enylmorpholine are mainly used in tobacco black shank control, but this results in a series of problems, such as high pesticide residues, environmental pollution and pathogen resistance to fungicides [3]. Therefore, it is of great significance to develop efficient, safe and eco-friendly biocontrol measures.

Prickly ash (*Zanthoxylum bungeanum*, Rutaceae) is an important economic crop species in China, with an annual yield of 100 million hectares. In recent years, this species has received increasing attention due to its economic significance (important condiment and traditional Chinese medicine) and ecological value (ideal plant for afforestation in barren

hills). Dried and peeled seeds of prickly ash (PAS) are the main by-products of the prickly ash industry. Nearly 1.5 billion kilograms of PAS are discarded as agricultural waste every year, resulting in resource waste and environmental pollution. In fact, PAS are rich in nutrients and bioactive components. Previous studies have shown the potential of PAS as a fumigant for controlling plant pathogens and root-knot nematodes [4,5]. However, it is unclear if PAS is a suitable material for bio-organic fertilizer (BOF) production.

It is well known that biochar addition can effectively prevent greenhouse gas emissions and nutrient loss during compost fermentation [6]. It also can regulate the growth of microorganisms and sequester carbon [7]. In addition, the application of an organic fertilizer with biochar could improve soil health, increase crop yields [8] and induce plant disease resistance [9]. We propose that biochar can also promote BOF production using PAS.

Currently, the use of BOF is considered to be an effective way to promote plant growth, inhibit disease occurrence and regulate soil microbial community structure [10]. For example, the combined use of *Bacillus amyloliquefaciens* W19 with organic fertilizer can decrease the incidence of *Fusarium* wilt in banana [11]. Our previous study revealed that PAS can inhibit a variety of microorganisms in the soil, but it increases *Bacillus* sp. [4], suggesting that it is feasible to produce *Bacillus* biocontrol agents by PAS. Therefore, in the current study, a novel BOF, named PBB, was produced from PAS with *Bacillus subtilis* Tpb55 inoculation and biochar addition, and its potential on tobacco black shank control was also evaluated.

## 2. Materials and Methods

### 2.1. Materials

*Phytophthora nicotianae* (*Pn*) strain JM01 was isolated and preserved by the Tobacco Research Institute of Chinese Academy of Agricultural Sciences. It was cultured in oatmeal agar medium (OA). *Nicotiana tabacum* 'Xiaohuangjin 1025', which is susceptible to tobacco black shank disease, was chosen for the experiment. PAS of *Zanthoxylum bungeanum*, commonly called Dahongpao, was chosen for the experiment because it is the main cultivated variety of this species in Shandong Province. Biochar was prepared from rice husks by anoxic pyrolysis at 400 °C. The air-dried moisture content (MAD), air-dried ash (ADA) and free radical volatiles (VAD) of this biochar were 3.16% ($w/w$), 14.34% ($w/w$) and 5.77% ($w/w$), respectively.

### 2.2. Optimization of PBB Production in Solid State Fermentation

The orthogonal experiment was designed to optimize the medium formulation for solid-state fermentation. PAS, biochar and soybean powder were mixed in proportion (Table 1), placed into an Erlenmeyer flask (100 g/1 L) and sterilized at 121 °C for 20 min. Then, *B. subtilis* Tpb55 suspension ($10^8$ cfu/mL) was inoculated (2.5 mL/100 g), and the final water content of the medium was adjusted to 50% with sterile water. The fermentation was carried out at 28 °C in the dark, and the flask was shaken once by hand every 12 h. The pH and bacteria population were determined once a day and successively for one week. The key factors affecting the fermentation were obtained by orthogonal analysis of the data on the fourth day. Each treatment was repeated three times.

### 2.3. Chemical Property Determination of PBB

PBB samples were dried in an oven at 60 °C for 24 h, and the ratio of the mass difference before and after drying to the mass of the sample before drying was the water content of the sample. The properties including pH, N content, C sequestration content, available P, available K, ammonium N and nitrate N were determined according to the method described by Bao [12].

**Table 1.** The results of PAS fermentation with *B. subtilis* inoculation and biochar addition.

| Number | Orthogonal Treatment Design | | | pH (Day 0) | pH (Day 7) | Colony Density [c] (cfu/g) |
|---|---|---|---|---|---|---|
| | PAS [a] (%) | >SP [b] (%) | >Biochar (%) | | | |
| 1 | 50 | 15 | 15 | $6.33 \pm 0.00$ e | $6.82 \pm 0.01$ d | $8 \times 10^8$ d |
| 2 | 50 | 10 | 30 | $6.54 \pm 0.00$ b | $6.93 \pm 0.01$ b | $5 \times 10^8$ d |
| 3 | 50 | 5 | 45 | $6.65 \pm 0.01$ a | $6.96 \pm 0.01$ b | $1.7 \times 10^{10}$ a |
| 4 | 60 | 15 | 45 | $6.64 \pm 0.00$ a | $7.05 \pm 0.01$ a | $4.1 \times 10^9$ b |
| 5 | 60 | 10 | 15 | $6.39 \pm 0.01$ de | $7.02 \pm 0.02$ a | $5.9 \times 10^9$ b |
| 6 | 60 | 5 | 30 | $6.55 \pm 0.00$ b | $6.93 \pm 0.00$ b | $2.8 \times 10^9$ c |
| 7 | 70 | 15 | 30 | $6.52 \pm 0.00$ bc | $6.78 \pm 0.02$ e | $4.4 \times 10^9$ b |
| 8 | 70 | 10 | 15 | $6.37 \pm 0.00$ de | $6.87 \pm 0.00$ c | $2.4 \times 10^9$ c |
| 9 | 70 | 5 | 45 | $6.45 \pm 0.09$ cd | $6.89 \pm 0.02$ c | $1.6 \times 10^{10}$ a |
| 10 | 0 | 100 | 0 | $6.38 \pm 0.02$ de | $6.12 \pm 0.02$ g | $1.4 \times 10^{10}$ a |
| 11 | 100 | 0 | 0 | $6.13 \pm 0.01$ f | $6.17 \pm 0.01$ f | $1.8 \times 10^4$ e |

[a], prickly ash seeds; [b], soybean powder; [c], the colony density of Tpb55 on the fourth day of fermentation. Different letters within a column indicate significant differences among biochar-amended soils at $p < 0.05$.

### 2.4. Seed Germination Test for PBB

We added distilled water to the PBB at a ratio of 10 g: 100 mL, soaked the mixture for 3 h (150 rpm) and centrifuged it for 10 min (3000 rpm). We then poured 10 mL of supernatant into the culture dish with filter paper, placed 30 rape seeds on the filter paper and incubated them at 21 °C for 48 h. An equal volume of distilled water served as a control. We recorded the percentage of seed germination, and a germination rate of more than 90% was considered indicative of sufficient fermentation.

### 2.5. The Fumigation Inhibition Effect of PBB on P. nicotianae

A mycelial disc was inoculated with *Pn* at the center of an oat medium plate. The plate was inverted and 4 g of PBB was evenly spread to cover the culture dish. The plate was then sealed and cultured at 25 °C for four days. Treatment with no PBB addition served as a control. Each treatment was repeated three times.

### 2.6. Pot Experiment

The inocula of *Pn* were prepared according to the method in reference [4]. Briefly, millet was boiled until 2/3 of the husks were broken, after which it was filtered through gauze, air-dried to approximately 40% water content and sterilized in a conical flask (121 °C, 20 min). *P. nicotianae* was activated on the OA medium for seven days. The obtained fungus cakes were inoculated in a sterilized millet medium (5 mm in diameter, three fungus cakes per bottle) with a hole punch and cultured at 26 °C for 14 days.

The tested soil was collected from the tobacco field, and it was mixed with the *Pn* cultured by millet (4 g millet/1 kg of soil) to obtain the diseased soil. There were four treatments in the experiments, i.e., 2% PBB addition (PBB2), 4% PBB addition (PBB4) and no PBB addition (control). Each pot was concealed with a plastic film for 10 days before transplanting. There were 15 replicates per treatment, and each treatment was repeated three times. The incidence of disease was assessed on the 15th day after inoculation, and the disease index, incidence rate and disease prevention effect were statistically analyzed [13].

### 2.7. Real-Time Fluorescent Quantitative PCR of P. nicotianae

On the 5th and 15th day after potting, rhizosphere soil of the tobacco seedlings was sampled, and the population of *P. nicotianae* was detected by real-time fluorescent quantitative PCR (RT qPCR). According to the 18S rDNA gene sequence of *P. nicotianae* from the NCBI database, the following specific primers were designed for fluorescent quantitative PCR: SP: 5′-TGAAGAACGCTGCGAACTGC-3′, AP: 5′-CTGACATCTCCTCCACCGACTA-3′. The length of the amplified target fragment was 172 bp [4]. DNA was extracted and purified from rhizosphere soil by a DNeasy®PowerSoil®Kit ((Qiagen, Hilden, Germany).

The RT-qPCR reaction system had a total volume of 20 μL, consisting of 2.0 μL cDNA, 10.0 μL SYBR premix (Takara, Japan), 0.4 μL SP (10 μM/μL), 0.4 μL AP (10 μM/μL), 0.4 μL ROX Reference DyeII (50×) and 6.8 μL ddH$_2$O. The amplification procedure was as follows: 94 °C for 5 min, 94 °C for 20 s, 65 °C for 40 s, and 72 °C for 40 s, with a total of 40 cycles.

### 2.8. Soil Chemical Property Determination

On the 20th day after potting, soil chemical properties were determined according to the method described by Bao [12]. The content of alkaline hydrolyzed N (AN) in the soil was determined by the alkaline hydrolyzed diffusion method; the content of available P (AP) in the soil was determined by the NaHCO$_3$ molybdenum–antimony anti-colorimetric method; the content of AK in the soil was determined by NH$_4$OAC torch photometry; and the content of organic matter was determined by the potassium dichromate volumetric method.

### 2.9. Microbial Diversity and Community of Rhizospheric Soil

On the 15th day after treatment, the rhizospheric soils of tobacco seedlings treated with 4% PBB and no PBB addition were collected. The total soil DNA was amplified by V3 + V4 regions of bacterial 16S rRNA genes (515F and 806R) and ITS1 regions of fungal 18S rRNA genes (ITS5-1737F and ITS2-2043R). Operational taxonomic units (OTUs) are defined as sequences with more than 97% similarity. α-diversity (Chao1, Shannon and ACE indexes) was calculated using Qiime (version 1.7.0), and β-diversity was analyzed by principal coordinate analysis (PCoA, Bray–Curtis).

### 2.10. Statistical Analysis

Microsoft Office Excel 2016 was used for data processing; IBM SPSS Statistics version 24.0 was used for variance and correlation analysis. The results of three replicates were expressed as means ± standard deviations. Variance analysis and multiple comparisons were used for determining statistical significance, and $p < 0.05$ was considered to indicate a significant difference. Principal coordinate analysis (PCoA) based on Bray–Curtis distance was used to compare the differences in bacterial and fungal community structure between the BOF and non-BOF treatment. Canonical correspondence analysis (CCA) was used to analyze the relationship between soil environmental factors and microbial community composition. PCoA and CCA were performed using R-2.15.3 for Windows.

## 3. Results

### 3.1. PBB in Solid-State Fermentation

The population of *B. subtilis* Tpb55 increased linearly in the first three days of fermentation and then logarithmically at the fourth day, resulting in saturation (Figure S1). Treatment 3 (10:1:9 of PAS: soybean cake: biochar) and treatment 9 (5:3:9 of PAS: soybean powder: biochar) showed the highest fermentation effect, with a colony density of 1.6–1.7 × 10$^{11}$ cfu/g, which is equal to that of soybean cake powder alone (Table 1). The fermentation with PAS alone showed the lowest colony density of Tpb55, indicating that the fermentation process can be greatly influenced by biochar addition. This was further confirmed by the results of orthogonal analysis (Table S1), which identified biochar as the most significant influencing factor for fermentation ($p < 0.0001$, R$^2$ = 0.831).

During the fermentation, the pH value varied in the different treatments. On the fourth day, the pH values of treatments with biochar addition all increased significantly, with a pH closer to 7 (Table 1). No significant pH change was observed in samples treated with PAS fermentation alone, while the pH of soybean cake powder alone decreased significantly. These results indicated that biochar addition not only can promote fermentation but also increase the pH of fermentation.

### 3.2. Biochemical Characteristics of PBB

Based on the fermentation formula optimization results, 10:1:9 of PAS: soybean cake: biochar was chosen as the fermentative material to produce PBB, and the fermentation time was 4 days. The germination rate of rape seeds treated with BOC extract was more than 90%, indicating that the fermentation reached the primary maturities. The main biochemical characteristics of BOC were as follows: bacteria $1.7 \times 10^{11}$ cfu/g, pH 6.97, organic carbon 35.64%, alkaline hydrolyzed N 11.2 g/kg, available P 21.69 mg/kg and available K 7574 mg/kg.

### 3.3. Inhibition Effect of PBB on P. nicotianae

PBB showed a good fumigation effect on *P. nicotianae* mycelium growth, with an inhibition rate of 76.67% when treated with a volume of 4 g/dish. As shown in Figure 1, the hyphae fumigated with BOF were limited to a very small area, and aerial mycelia grew loosely, while the control grew vigorously and almost covered the entire dish.

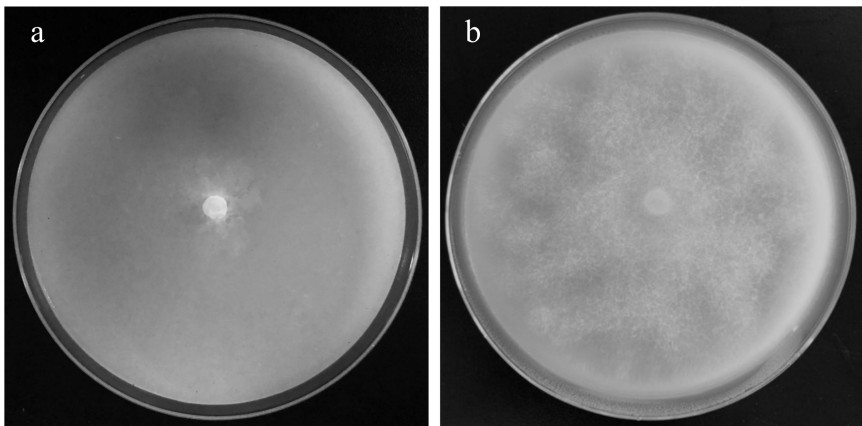

**Figure 1.** Fumigation effect of PBB on *Phytophthora nicotianae*. (**a**), 4 g/dish of PBB; (**b**), control.

As shown in Table 2, BOF can also influence the survival of *Pn* in soil. Compared to the control, DNA copies of *Pn* with 2% and 4% of BOF addition decreased 79.98% (PBB2) and 99.73% (PBB4) after five days of inoculation, respectively. Moreover, the inhibitory effect was sustained for 15 days. The inhibition rates of 2% and 4% BOF addition reached 98.94% and 99.76% on the fifteenth day.

**Table 2.** The effect of PBB treatment on *P. nicotianae* population and disease severity.

| Treatment | DNA Copies of *P. nicotianae* | | Disease Index |
|---|---|---|---|
| | **Fifth Day** | **Fifteenth Day** | **Fifteenth Day** |
| PBB4 | 3494.31 ± 85.77 d | 3346.10 ± 95.92 d | 25.93 ± 2.21 c |
| PBB2 | 277,980.00 ± 1346.43 c | 14,717.10 ± 174.78 d | 40.74 ± 2.45 b |
| CK | 1,312,089.77 ± 5765.04 a | 1,388,622.37 ± 12,616.09 a | 63.89 ± 2.07 a |

PBB4 and PBB2 represent 2% and 4% of bio-organic fertilizer addition, respectively; control, with no fertilizer addition. Different letters within a column indicate significant differences among biochar-amended soils at $p < 0.05$.

### 3.4. Prevention Effect of PBB Application on Tobacco Black Shank Disease

The pot experiment showed that bio-organic fertilizer addition could alleviate the severity of tobacco black shank (Table 2). The disease index of treatment with BOF addition was lower than that in the control ($p < 0\,05$). Moreover, 4% BOF addition showed a better preventive effect (59.41%) than 2% PBB addition (36.23%).

### 3.5. Soil Chemical Properties Affected by PBB Addition

As shown in Table 3, PBB application significantly increased the soil pH, hydrolysable N, available K and organic carbon, but it decreased the electrical conductivity compared to the control. The content of available P had no significant difference between treatment and control. Except for the increase in pH and available K content, the two BOF addition levels had no significant effect on the main soil nutrient supply, including available P, hydrolysable N and organic matter.

**Table 3.** Effects of bio-organic fertilizer treatment on soil chemical properties.

| Treatment | pH | Electrical Conductivity (μs/cm) | Available P (mg/kg) | Available K (mg/kg) | Hydrolysable N (mg/kg) | Organic Carbon (g/kg) |
|---|---|---|---|---|---|---|
| PBB4 | 6.79 ± 0.02 a | 677.00 ± 4.36 c | 72.76 ± 1.41 b | 847.31 ± 3.30 a | 357.39 ± 5.66 a | 102.09 ± 0.39 a |
| PBB2 | 6.56 ± 0.00 b | 793.00 ± 12.17 b | 80.86 ± 1.29 a | 696.60 ± 3.35 b | 301.52 ± 38.82 b | 102.77 ± 2.29 a |
| CK | 6.19 ± 0.03 c | 1148.33 ± 103.00 a | 72.75 ± 0.16 b | 462.22 ± 3.31 c | 288.63 ± 10.90 b | 69.82 ± 3.57 b |

PBB4 represents 4% PBB addition; control, with no PBB addition. Different letters within a column indicate significant differences among biochar-amended soils at $p < 0.05$.

### 3.6. Microbial Diversity and Richness of Rhizospheric Soil

The alpha diversity analysis revealed that the fungal diversity and richness index with PBB addition were lower than that of the control. No significant difference in bacterial diversity or richness index was observed between PBB and the control. These results indicated that fungi were more susceptible to PBB treatment than bacteria (Table 4).

**Table 4.** Operational taxonomic unit (OTU) richness and diversity indices of PBB treatment.

| Treatment | Fungal Community | | | | Bacterial Community | | | |
|---|---|---|---|---|---|---|---|---|
| | OTUs | Shannon | Chao1 | ACE | OTUs | Shannon | Chao1 | ACE |
| Control | 207 ± 11 a | 2.78 ± 0.09 a | 235 ± 12 a | 237 ± 11 a | 1474 ± 119 a | 7.50 ± 0.28 a | 1612 ± 110 a | 1637 ± 105 a |
| PBB | 138 ± 6 b | 2.53 ± 0.07 b | 160 ± 15 b | 177 ± 20 b | 1362 ± 38ab | 7.07 ± 0.27 a | 1517 ± 24 a | 1534 ± 26 a |

PBB, 4% bio-organic fertilizer addition; control, no PBB addition; Different letters within a column indicate significant differences among biochar-amended soils at $p < 0.05$.

### 3.7. Effect of BOF Addition on Soil Microbial Community Structure

Based on the T-test, the relative abundance of nine fungal genera and nine bacteria genera displayed significant differences between PPB treatment and the control (Table 5). PBB addition greatly increased the richness of *Mucor*, *Rhizopus*, *Cunninghamella*, *Chitinophaga*, *Phenylobacterium* and *Bacillus*. Nine genera were significantly decreased, including *Penicillium*, *Trichoderma*, *Fusarium*, *Serendipita*, *Arthrobotrys*, *Helvella*, *Sphingomonas*, *Rhodanobacter* and *Gemmatimonas*. In addition, the abundance of *Bacillus* in the treatment with PBB addition was significantly higher than that in the control (Table 5), which was consistent with the expected results of fermentation with *B. subtilis* Tpb55 inoculation.

PCoA showed the relationship between the compositions of all soil microbial communities (Figure 2). As shown in Figure 2, the four repeats randomly collected from each treatment had similar positions and were located in different quadrants. Among them, the bacterial community (Figure 2a) had a contribution of 63.81% on the abscissa and 34.43% on the ordinate, and the fungal community (Figure 2b) had a contribution value of 70.62% on the abscissa and 13.23% on the ordinate. The ADONIS test (Table S1) showed that the bacterial ($R^2 = 0.6162$, $p = 0.03$) and fungal ($R^2 = 0.6502$, $p = 0.031$) communities of PBB and control were significantly different. In summary, the composition of the soil microbial community changed significantly with the PBB application.

**Table 5.** Genera with significant differences in relative abundance between BOF addition and control.

| Category | Genus | Relative Abundance (%) | |
| --- | --- | --- | --- |
| | | **4% PBB Addition** | **Control** |
| Fungi | *Mucor* | 28.8 ± 3.60 a | 2.83 ± 0.55 b |
| | *Rhizopus* | 30.41 ± 2.26 a | 0. 76 ± 0.12 b |
| | *Penicillium* | 3.41 ± 0.62 b | 23.13 ± 1.27 a |
| | *Trichoderma* | 2.32 ± 0.40 b | 6.94 ± 0.66 a |
| | *Fusarium* | 0.12 ± 0.03 b | 0.79 ± 0.08 a |
| | *Cunninghamella* | 0.41 ± 0.001 a | 0.00 ± 0. 01 b |
| | *Serendipita* | 0.00 ± 0.00 b | 0.11 ± 0.01 a |
| | *Arthrobotrys* | 0.00 ± 0.00 b | 0.08 ± 0.002 a |
| | *Helvella* | 0.00 ± 0.001 b | 0.03 ± 0.0001 a |
| Bacteria | *Chitinophaga* | 11.77 ± 0.03 a | 1.95 ± 0.39 b |
| | *Sphingomonas* | 0.04 ± 0.002 b | 0.09 ± 0.003 a |
| | *Chitinophagaceae* | 1.14 ± 0.03 b | 0.06 ± 0.01 a |
| | *Dyella* | 2.54 ± 0.01 a | 1.93 ± 0.12 b |
| | *Rhodanobacter* | 1.24 ± 0.31 b | 4.24 ± 0.43 a |
| | *Phenylobacterium* | 3.51 ± 0.73 a | 1.64 ± 0.175 b |
| | *Bacillus* | 4.07 ± 0.43 a | 2.86 ± 0.24 b |
| | *Gemmatimonas* | 0.89 ± 0.16 b | 2.81 ± 0.72 a |

PBB, 4% bio-organic fertilizer addition; control, no PBB addition; Different letters within a column indicate significant differences among biochar-amended soils at $p < 0.05$.

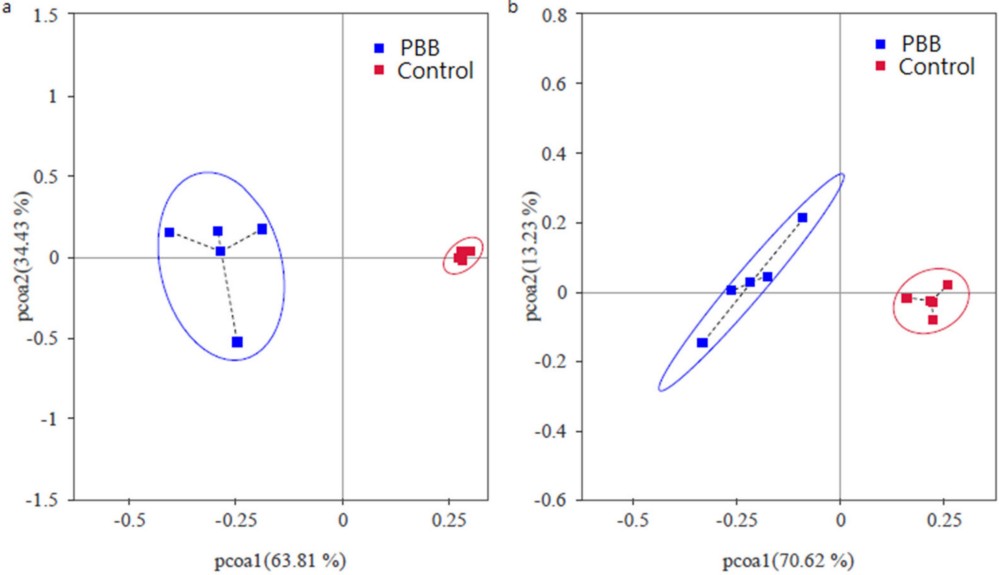

**Figure 2.** Principal coordinate analysis (PCoA) of soil fungal (**a**) and bacterial (**b**) communities based on Bray–Curtis distance. PBB represents 4% of PBB addition; control, with no PBB addition.

*3.8. Correlation between Soil Microbial Community and Soil Environmental Factors*

CCA analysis was used to further identify the soil characteristics that could contribute to microbial community variations affected by PBB addition. Both fungi and bacteria communities with BOF addition were well-separated from those in samples without PBB treatment (control). The length of the arrow represents the correlation. It can be seen from Figure 3 that the distribution of the fungi community from PBB samples was significantly positively correlated with pH, OC, N and K, while that from non-PBB samples was related to EC (Figure 3a). A similar tendency was also found in the bacterial community (Figure 3b). At the genus level, *Cunninghamella, Mucor, Rhizopus, Achromobacter, Chitinophaga, Phenylobacterium* and *Ochrobactrum* were positively correlated with pH, OC, TN and AP, while the main influencing factor for *Trichaoderma, Actinomucor, Fusasium, Penicillium, Arthrobotrys, Serendipita, Sphingomonas, Rhodanobacter* and *Gemmatimonas* was EC (Figure 3).

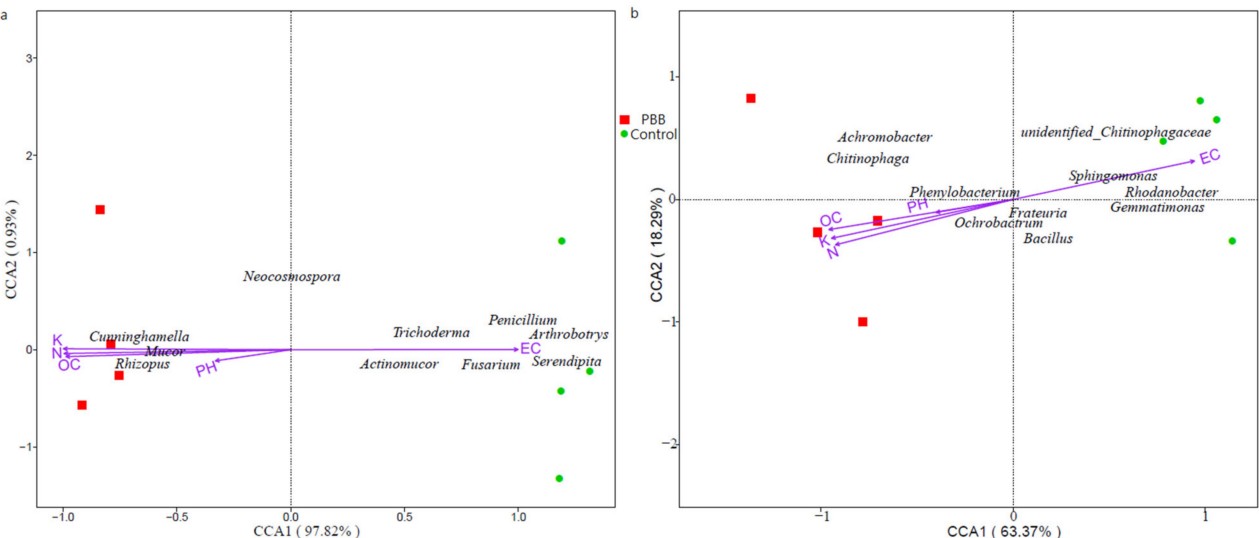

**Figure 3.** Canonical correspondence analysis (CCA) diagram illustrating the relationship between fungal (**a**) and bacterial (**b**) composition at genus level from different samples and environmental variables.

According to the partial Mantel test (Table S2), available potassium and EC had the most significant effect on soil fungi community structure ($R^2$ = 0.9939, $p < 0.01$ and $R^2$ = 0.9937, $p$ = 0.01), followed by AN ($R^2$ = 0.9872, $p$ = 0.0225) and OC ($R^2$ = 0.9719, $p$ = 0.0285). The soil bacterial community was affected significantly by AK ($R^2$ = 0.9833, $p < 0.01$) and total N ($R^2$ = 0.9837, $p < 0.01$), followed by OC and EC. pH showed no significant effect on either the fungal or bacterial community. Therefore, EC, available potassium, TN, OC and EC may be important environmental factors affecting soil microorganisms.

## 4. Discussion

BOF produced by antagonistic microbes can effectively control soil-borne diseases of tobacco, cucumber, watermelon and other crops [14–16]. Agricultural straw, livestock manure and soybean meal are conventional raw materials for BOF production, while PAS is rarely used. The current study showed that using PAS as a substitute for soybean powder is feasible for *B. subtilis* Tpb55 production. The final number of bacteria in PBB was more than $10^{10}$ cfu/mL under the optimal PAS fermentation formula, which is equal to that produced by 100% of soybean powders. However, the growth of the Tpb55 strain was very low in PAS with no biochar addition, indicating that biochar has a great influence on the fermentation process. The promotion of biochar in microbial fermentation has been widely reported. The promotional effect can be associated with the pore structure and low density of biochar, which can increase the oxygen flux of the fermentation substrate [6,17]. It is especially important for the fermentation of PAS, harboring high fat content and low air permeability. Alleviating the pH decline may be another important role of biochar in fermentation. Additionally, the biochar also significantly increased the final pH of PBB. To the best of our knowledge, this is the first report to produce BOF by PAS with biochar addition and *B. subtilis* inoculation.

BOF is considered to have advantages of both organic fertilizers and antagonistic microbes. Organic fertilizers can provide nutrients for antagonistic microbes and promote their colonization in soil [18,19]. These nutrients are also beneficial for plant growth [20,21]. Chen et al. [22] found that BOF fortified with Bacillus licheniformis X-1 and Bacillus methylotrophicus Z-1 showed a significant preventive effect against strawberry Fusarium wilt disease. Hafez et al. [23] reported that a BOF produced from spent grain waste with *Azospirillium brasilense* inoculation enhanced soil fertility and corn growth. Consitantly, PBB showed inhibitory activity against *P. nicotianae*, reduced the survival of pathogens in soil and decreased the disease severity of tobacco black shank. The soil pH, OC, AN and AK were also increased with PBB application. These results indicated for alkaline

hydrolyzed N that pathogen inhibition and soil nutrient improvement were involved in the effects of PBB.

Soil microbial community structure not only affects plant growth but also has a close relationship with plant diseases [24,25]. The ability of BOF in the regulation of the soil microbial community has been widely reported, and they are considered an important disease prevention strategy [26,27]. Shen et al. [27] reported that BOF treatment shifted the soil microbial community and enhanced the populations of some beneficial microorganisms, such as *Gemmatimonas* and *Sphingomonas*. Zhao et al. [16] found that the application of BOF enriched with *B. amyloliquefaciens* increased the bacterial diversity but decreased the fungal diversity in soil, and populations of some potential beneficial taxa such as Firmicutes (*Bacillus*) and Basidiomycota were enhanced. In line with these results, PBB treatment reduced the richness and diversity of the fungal community but had no significant effect on that of bacteria. PBB application also altered the microbial structure in tobacco rhizospheric soil. As expected, the relative abundance of *Bacillus* increased with PBB treatment. In addition, some potential beneficial genera were also enriched, including *Mucor, Rhizopus, Cunninghamella, Chitinophaga* and *Phenylobacterium* [28–32]. Nine genera were significantly decreased, including *Fusarium*, which includes pathogens that can cause a variety of crop diseases [15,16]. These results indicate that organic fertilizer combined with biocontrol bacteria is more conducive to the regulation of the soil microecological environment, thus playing a dual role in promoting growth and preventing disease.

## 5. Conclusions

This study demonstrated that the agricultural waste PAS combined with biochar can replace soybean as a source for the production of biocontrol *B. subtilis* Tpb55. Pathogen inhibition, soil nutrient improvement and rhizomicrobial community shifting are included among the effects of the novel BOF for tobacco black shank control. However, the effect of PBB on *B. subtilis* Tpb55 colonization in rhizosphere soil needs further study.

**Supplementary Materials:** The following are available online at https://www.mdpi.com/article/10.3390/app11167261/s1, Figure S1: The changes in bacteria number with different fermentation formulas, Table S1: ADONIS analysis using Bray–Curtis distance between PBB treatment and the control, Table S2: Partial Mantel test for whole OTUs of soil microbe versus environmental factors.

**Author Contributions:** Conceptualization, C.Z. and X.H.; investigation, X.Z. and Y.W.; writing—original draft preparation, X.Z.; funding acquisition and project administration, J.G.; methodology, W.L.; writing—review and editing, C.Z. All authors have read and agreed to the published version of the manuscript.

**Funding:** This research was funded by the China National Tobacco Corporation (110201902003) and the Science and Technology Project of Guizhou Tobacco Corporation (201809).

**Conflicts of Interest:** The authors declare no conflict of interest.

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
