# Peer review of "A Novel Bio-Fertilizer Produced by Prickly Ash Seeds with Biochar Addition Induces Soil Suppressiveness against Black Shank Disease on Tobacco"

_applsci, doi:10.3390/app11167261_

Round 1

Reviewer 1 Report

This manuscript demonstrates the effect of a biological organic fertilizer (BOF) produced by prickly ash seeds (PAS) on tobacco black shanks. In combination with biocontrol bacteria, the treatment of BOF decreased fungal diversity in soil, whereas it increased some beneficial microbial community. The usefulness of eco-friendly BOF and the new role of PAS are interesting. I think this manuscript deserves to be published in this journal after some revisions.

1. The authors mentioned "The promotion effect can be associated to the pore structure and low density of biochar, which can increase the oxygen flux of fermentation substrate." in the discussion. Please provide any evidence to support this statement. 

2. It is not clear what PBB stands for.

3. in line 300; groth --> growth

Author Response

Dear Editor and reviewer,

Thanks very much for you and the anonymous reviewer. We have revised our manuscript carefully according to your and the referee's comments. Our responds to questions are as follows. Any other questions, please contact us. Thanks again for your effort for our papers.

To Review’s Comments:

  1. Question: The authors mentioned "The promotion effect can be associated to the pore structure and low density of biochar, which can increase the oxygen flux of fermentation substrate." in the discussion. Please provide any evidence to support this statement.

Answer: Thanks for your insightful comment. We have supplemented reference supporting this statement (line 284). It was stated in references that the high porosity and low density of biochar could improve composting aeration, thus enhancing microbial activities and the humification process.

  1. Question: It is not clear what PBB stands for.

Answer: Thanks for your comment. There is an explanation in Lines 64-66 that “In the current study, a novel BOF, named PBB, was produced by PAS with Bacillus subtilis Tpb55 inoculation and biochar addition.” That is to say, PBB is short for PAS fertiliser fermented with biochar and Bacillus subtilis.

  1. Question: in line 300; groth --> growth

Answer: Thanks for your comment. We have revised this error.

Once again, thank you very much for your comments and suggestions.

With Best Regards.

Sincerely Yours

Cheng-Sheng Zhang

Reviewer 2 Report

Dear Authors,

In the presented manuscript a  novel bio-fertilizer, produced by prickly ash seeds (PAS), Bacillus subtilis and biochar, was evaluated for its disease preventing potential on tobacco black shank caused by Phytophthora  nicotianae.

Research is gripping as well as it has a scientific value. The results of the study provided useful and practical information that could be effective in the reducing of economic losses in many important crops of some plants such as citrus, tobacco etc.

The introduction provides a good, generalized background of the topic that quickly gives the reader an  appreciation of the scientific relevance and timeliness of the research theme.  I think the motivations for this study are very clear. The experimental apparatus is quite standard, and is appropriate for the study. The data didn’t duplicated in the graphics and/ text. The figures and tables are an  easy to interpret. An appropriate statistical methods have been used to test the significance of the results. The findings are  described in the context of the published literature. The conclusions of the study were supported by appropriate evidence.

However, there are some  minor flaws in manuscript what need to be fixed. Specific comments on the manuscript are as follows:

Line 78: Please, complete what kind of % was used? (w/w, v/w, v/v..)

Please, complete information about PCoA (Figure 2 )  and CCA (Figure 3) in subsection 2.10 (Lines: 156-161).

The whole manuscript seems gripping and is well-written. In my opinion, this manuscript is appropriate for publication in Journal Applied Sciences, after minor revision. 

Author Response

Dear Editor and reviewer,

Thanks very much for you and the anonymous reviewer. We have revised our manuscript carefully according to your and the referee's comments. Our responds to questions are as follows. Any other questions, please contact us. Thanks again for your effort for our papers.

To Review’s Comments:

  1. Question: Line 78: Please, complete what kind of % was used? (w/w, v/w, v/v..)

Answer: Thanks for your comment. We have completed this information. All of those % were w/w.

  1. Question: Please, complete information about PCoA (Figure 2) and CCA (Figure 3) in subsection 2.10 (Lines: 156-161).

Answer: Thanks for your comments. We have completed this information (lines: 154-158). Principal coordinate analysis (PCoA) based on bray-curtis distance was used to compare the differences of bacterial and fungal community structure between the BOF and non-BOF treatment. Canonical correspondence analysis (CCA) was used to analyse the relationship between soil environmental factors and microbial community composition. PCoA and CCA were performed using R‐2.15.3 for Windows.

Once again, thank you very much for your comments and suggestions.

With Best Regards.

Sincerely Yours

Cheng-Sheng Zhang